# Removal of an Ethoxylated Alkylphenol by Adsorption on Zeolites and Photocatalysis with TiO₂/Ag

**Claudia Aguilar Ucán [1],\*** , **Mohamed Abatal [2]** , **Carlos Montalvo Romero [1]**,
**Francisco Anguebes Franseschi [1]** , **Miguel Angel Ramirez Elias [1]** and **Denis Cantú Lozano [3]**

[1] Faculty of Chemical Sciences, Autonomous University of Carmen, Street 56 No. 4, Carmen City, Campeche C.P 24180, Mexico; c_montalvo10@hotmail.com (C.M.R.); fanguebes@pampano.unacar.mx (F.A.F.); mramirez@pampano.unacar.mx (M.A.R.E.)

[2] Faculty of Engineering, Autonomous University of Carmen, Central Avenue S/N, Maya World, Carmen City, Campeche C.P 24115, Mexico; mabatal@pampano.unacar.mx

[3] National Tecnologist of México, East 9, Emiliano Zapata, Orizaba VER 94320, Mexico; dencantu@gmail.com

\* Correspondence: alejandra175@hotmail.com

**Abstract:** Two advanced removal methods (adsorption and photocatalysis) were compared for the elimination of an ethoxylated alkylphenol (nonylphenol polyethylene glycol, NPEG). For the adsorption process, zeolites were used in their natural state, and the process was characterized by DRX (X-ray diffraction) and SEM–EDS (Scanning electron microscopy). The analysis of the results of the adsorption kinetics was carried out using different isotherms to interpret the removal capacity of zeolites. The Temkin kinetic model better predicted the experimental data and was satisfactorily adjusted to models of pseudo-second order (PSO). On the other hand, for photocatalysis, nano-particles of Ag (silver) were deposited on titanium oxide (TiO₂) Degussa-P25 by photo-deposition, and the catalyst was characterized by diffuse reflectance and SEM–EDS. The data obtained using the two removal techniques were analyzed by UV–Vis (ultraviolet-visible spectrophotometry) and total organic carbon (TOC). The kinetic data were compared. The photocatalytic process showed the highest efficiency in the removal of NPEG, corresponding to >80%, while the efficiency of the adsorption process was <60%. This was attributed to the recalcitrant and surfactant nature of NPEG.

**Keywords:** water pollution; ethoxylated alkylphenol; heterogeneous photocatalysis; zeolite adsorption

## 1. Introduction

Emerging pollutants are drugs, endocrine disruptors, personal care products, surfactants, pesticides, among other substances, found in drinking and wastewater; likewise, their presence in sediments and soils has been detected as a result of their excessive use. Recent studies show the presence of 4-nonylphenol, 4-tert-octiphenol, and bisphenol A in sediments and waters. Although the risk factors for organisms are still considered low, the potential danger posed by these components should be considered [1], In addition, other studies [2,3] report concentrations of these components in different substrates, finding a direct relationship between the months with low rainfall and the concentration of these pollutants [4].

Alkylphenols (APs) and bisphenol a (BPA) are common products of the chemical industry, used as surfactants in industrial detergents. Although not the object of this study, it is important to mention that bisphenol A is a monomer employed in the manufacture of polymers such as polycarbonate and epoxy resins and as a precursor for the synthesis of tetrabromobisphenol, which is used for food

packaging and has caused concern because it is used in the manufacture of plastic bottles and food containers and may be ingested by people consuming soft drinks or food [5],. Similarly, alkylphenols have been used as surfactants in the production of non-ionic detergents and in the cosmetics and disinfectants industry.

The excellent colloidal properties of surfactants has also used in the purification, control and separation of substances of importance, [6] how non-metal nano-particles (NPs), and two-dimensional materials. Graphene oxide (GO) is a two-dimensional material with a hydrophobic carbon lattice functionalized with hydrophilic oxygen groups on the edges; its basal hydrophilic nature allows it to behave as a surfactant, stabilizing emulsions of oil in water [7].

The use of surfactants allows the efficient separation of platinum particles on an large scale; replacing centrifugation processes that limit the performance with the addition of a colloidal suspension of NPs, the elimination of particles of undesired size is optimize [8].

The floatability of seven plastics in the presence of alkyl ethoxylated nonionic surfactants was investigated. It was found that the floatability of these plastics decreased with the addition of a surfactant [9].

Nano/submicron particles can be activated by surfactants and aggregate at the air–water interface to generate and stabilize foams. Such systems have been applied extensively in the food, medicine, and cosmetics industries [10]. The use of surfactants is diverse; however, their presence in the environment as waste must be eliminated.

The annual production of alkylphenols has been estimated at 154,000 tons in the USA [11,12]. The most important members of this group are nonylphenol ethoxylate (NPE) and octylphenol ethoxylate. These substances undergo metabolic degradation in the environment and the lose ethylene oxide side chains, transforming into alkylphenols (4-n-octylphenol and 4-n-nonylphenol). Unlike most exogenous chemicals, which generally become less toxic with biodegradation, alkylphenols increase their toxicity during this process [12].

Nonylphenol (NP) and its ethoxylated derivatives have been included as priority pollutants by the Ministry of Health and Environment of Canada. The effects of non-ionic surfactants as active hormones and endocrine disruptors have been studied in both humans and animals, and a significant number of reports describe adverse effects on the reproductive function [13–16]. The biodegradation of ethoxylated alkylphenols (APEs) during the treatment of wastewater or its subsequent discharge into the environment can result in the formation of small ethoxy chains and in the production of hydrophobic metabolites, such as mono- and di-ethoxylated nonylphenol, nonylphenoxy ethoxy acetic acid, nonylphenoxy acetic acid, and nonylphenol, which is more recalcitrant and more toxic than the other precursors of ethoxylated nonylphenols (NPEOs) [17].

Although the toxicity of APEs is relatively low, their study has increased in recent years due to their slow biodegradation and generation of highly stable, toxic, and bio-recalcitrant secondary products, especially those with one or two ethoxylated groups, such as nonylphenols and octylphenols, which have the ability to mimic the natural functions of hormones and behave as endocrine disruptors [17]. Their presence in treatment plants that use microorganisms for the reduction of organic matter causes serious problems due to their ability to produce foams, decrease the capacity of oxygen transfer, and disturb primary sedimentation processes, thus making their biodegradation inefficient and incomplete.

Conventional treatments for the degradation of these relatively new pollutants are inadequate or deficient, so various technologies have been proposed for their removal, such as membrane-based processes, biotechnological methods, oriented adsorption, and advanced oxidation processes [18]. Photocatalytic processes have proven to be highly efficient in the removal of emerging pollutants. They are based on new and innovative reactors that include solar technologies coupled to photo-catalytic processes [19]; photocatalytic rotary reactors have also been studied in pilot scale [20].

Degradation assisted by immobilized biocatalytic systems has been used in the removal of bisphenols, among other compounds, and has shown efficiency in the degradation of a broad spectrum of pollutants [21]. The removal of some emerging pollutants such as drugs (diclofenac, carbamazepine,

amoxicillin) has been performed with electro-membrane bioreactors (MBR), which offer the advantages of small dimensions and low mud production [22].

While photocatalysis has proven to be an efficient process for the removal of organic molecules, "load recombination" decreases the process efficiency [23]. In this regard, catalyst doping is an efficient means to achieve the deposition of metallic species on the surface of a catalyst in order to change its electrical properties and increase its efficiency for photocatalytic processes. For such purposes, titanium oxide ($TiO_2$) is a support for metal ions. Its structure consists of small particles of nano-metric size with a large surface area on which metallic silver can be deposited. In addition to preventing recombination (by sequestration of the electrons from the valence band), it provides photo-generated holes for the photocatalytic reaction [24]. The addition of silver on different semiconductors and nanocomposites transforms their specific surface and improves their photocatalytic activity and the separation of the electron–hollow pair [25–28].

On the other hand, natural zeolites have been used as ion exchangers in the removal of ammonium ions from waste and drinking water. It is well known that the ion-exchange property of zeolites is due to the presence of tetrahedra of $[AlO_4]^-$ in the structure of zeolite. In other words, the tetrahedron of $[AlO_4]^-$ contains Lewis acid sites, i.e., cationic sites where cations can be exchanged. Clinoptilolite is the most studied zeolite as an ion exchange material and is used commercially in the treatment of industrial and municipal wastewater to reduce the concentration of ammoniacal nitrogen. The ionic exchange of ammonium on natural zeolites has been studied extensively. Likewise, various natural and modified zeolites have been tested for the removal of aromatic structures such as phenols and have proven to be excellent adsorbents [29,30]; zeolites also appear efficient for the elimination of nonylphenol, octylphenol, and their extoxylated derivatives in systems combined with coagulation and ozone [31].

Due to the broad spectrum of uses of these compounds and their adverse effects on human health mentioned above, their elimination from the environment should be enhanced. The objective of this work is the study of the reaction kinetics of chemical transformation and adsorption used for the elimination of nonylphenol PEG to advance our understanding of the underlying phenomena.

## 2. Materials and Methods

### 2.1. Photocatalysis

The catalyst used was titanium oxide Degussa P-25 (Evonik Degussa México S. A. de C.V., CDMX, México); 4-Nonylphenyl-polyethylene (CAS number: 9016-45-9) was used as a substrate Silver nitrate ($AgNO_3$) grade P.A. (Fisher Scientific UK Ltd, Loughborough, UK) was used as a precursor salt for silver ions. The separation of the solution catalyst was performed using 0.22 mm cellulose filters (MilliporeCorp, Billerica, MA, US). Surface morphological analysis was performed by the secondary electron method, and chemical analysis by energy dispersion spectroscopy (EDS), using a Dual-Beam Scanning Electron Microscope (FIB/SEM) FEI-Helios Nanolab 600 from National Laboratory of Research in Nanosciences and Nanotecnology (LINAN, San Luis Potosí, México)

To obtain an estimate of the bandwidth value ($E_g$), the catalysts were analyzed by UV-vis spectroscopy using a Shimadzu UV-2450 system, equipped with the ISR-2200 Integrating Sphere Attachment from the Autonomous University of San Luis Potosi, México.

### 2.2. Adsorption

A natural zeolite (clinoptilolite) obtained from a reservoir located in the town of San Luis Potosí, Mexico, was used. Before use, the adsorbent was subjected to a chemical pre-treatment to remove impurities and improve the adsorption performance. Initially, the natural zeolite was screened to obtain the desired particle size (1–2.5 mm), washed with deionized water to remove dust attached to the surface of the particles, filtered under vacuum to extract moisture, dried in a muffle furnace (Felisa, Jalisco, México) at a temperature of 80 °C for a period of 12 h, and stored in a container.

## 3. Results and Discussion

### 3.1. Doping of the Catalyst and Characterization of TiO$_2$/Ag

Photo-deposition methods are based on the fact that certain metal cations with appropriate redox potentials can be deposited on a support and reduced by photoelectrons created by the lighting of UV lamps [32].

To perform the process in a glass reactor, 0.5 g of TiO$_2$ (Degussa P25) was deposited, and 100 mL of deionized water was added. To obtain a homogeneous solution under continuous agitation, it was left one hour in the dark, then nitrogen (80 cm$^3$/min) (INFRA Corporation, Carmen City, México) was added, and the reaction was subjected to the radiation of 4 UV lamps at 365 nm for 4 h. At the end of the irradiation period, water was removed by vacuum filtration followed by drying at 100 °C and calcination at 550 °C. The amount of AgNO$_3$ was estimated based on a weight/weight ratio.

Figure 1 shows the modifications of the optical properties of titanium due to the presence of silver nanoparticles on the surface. A significant improvement in absorption due to surface plasmon resonance between 500 and 600 nm was achieved as shown, resulting from the interaction of the metal particles with the incident light. The improvement of the photocatalyst efficiency by the inclusion of metallic particles such as silver could be confirmed by Fermi energy levels which were displaced at values close to the bottom of the conduction band; the accumulation of electrons influences the plasmon absorption band, which can improve the photocatalytic activity of a material in the visible region [20,23] our results indicates that the bandwidth of the titanium modified with silver was 2.9 EV.

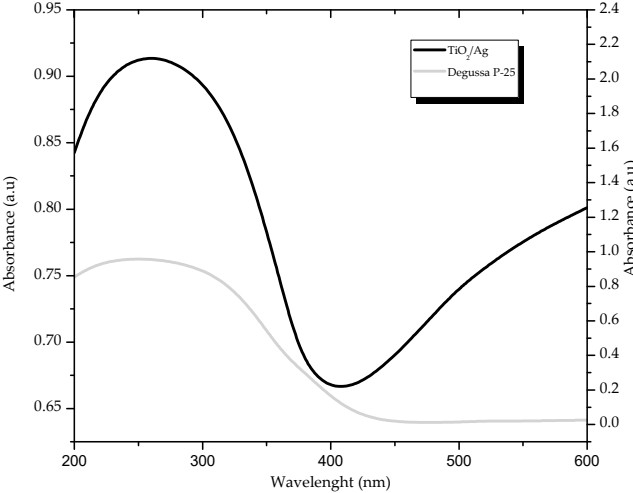

**Figure 1.** Comparison of the diffuse reflectance profiles of P-25 titanium and titanium doped with silver.

The results of X-ray diffraction (DRX), as well as the results of scanning electron microscopy (SEM) and those of the analysis of the chemical composition of the catalyst (SEM–EDS) have been published in a previous article [24]; they indicated that the actual percentage of deposited silver was 3.4% and that there was no evidence of alteration in the composition of the crystalline phase of the catalyst significantly due to the inclusion of metallic silver. According to Newbury [33]; these EDS values can be taken as an approximation of the real values due three factors: (1) the employment of standardless analysis, which is now commonly used in EDS quantitative analysis, (2) the effects of the specimen on the accuracy of X-ray microanalysis, and (3) incorrect elemental identification.

### 3.2. Clinoptilolite Characterization

Clinoptilolite was characterized in its natural state. The skeleton of the zeolite is composed of tetrahedrons of [SiO$_4$] and [AlO$_4$]$^-$, and the neutrality of the structure is preserved by balancing each tetrahedron with a positive charge provided by interchangeable cations (K$^+$, Na$^+$, Ca$^{2+}$ Mg$^{2+}$). Figure 2

shows the pattern of DRX, showing that the material was composed of calcium clinoptilolite and the following chemical elements: K 0.64, $H_2$ 0.423, Na 0.64, Ca 0.66, Mg 0.26, O 49.2, Si 18, indicating an aluminosilicate structure.

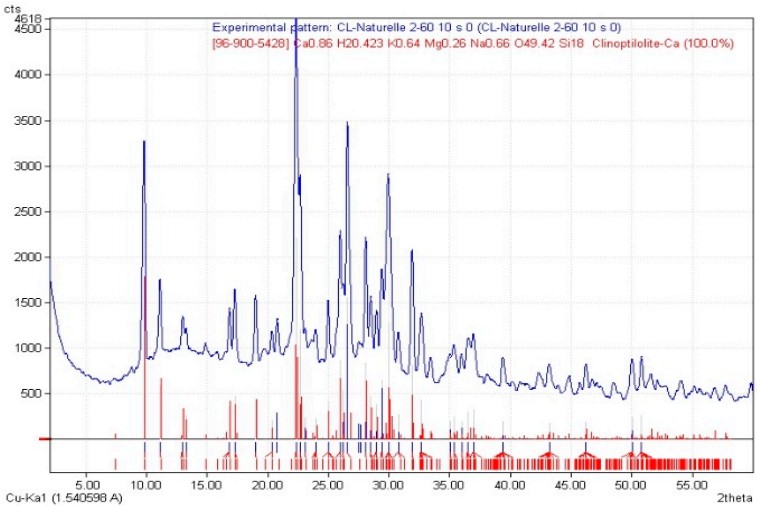

**Figure 2.** X-ray diffractogram of the zeolite clinoptilolite.

The SEM–EDS analysis (Figure 3) revealed the presence of aggregates of thin crystals of the order of 5.17 μm in the form of plates with different geometry. Their shape was consistent with that of the clinoptilolite zeolite. The chemical composition of the zeolite was determined by EDS analysis, obtaining Al = 3.85%, Si = 27.92%, which confirmed that the material was a micro-porous crystalline aluminum silicate formed by the skeletons of the $SiO_4$ and $AlO_4$ tetrahedrons. The values obtained for low-molecular-weight elements such as oxygen may not be correct when using EDS and are for qualitative purposes only. For elements of greater molecular weight such as Al and Si, EDS techniques are usually accurate.

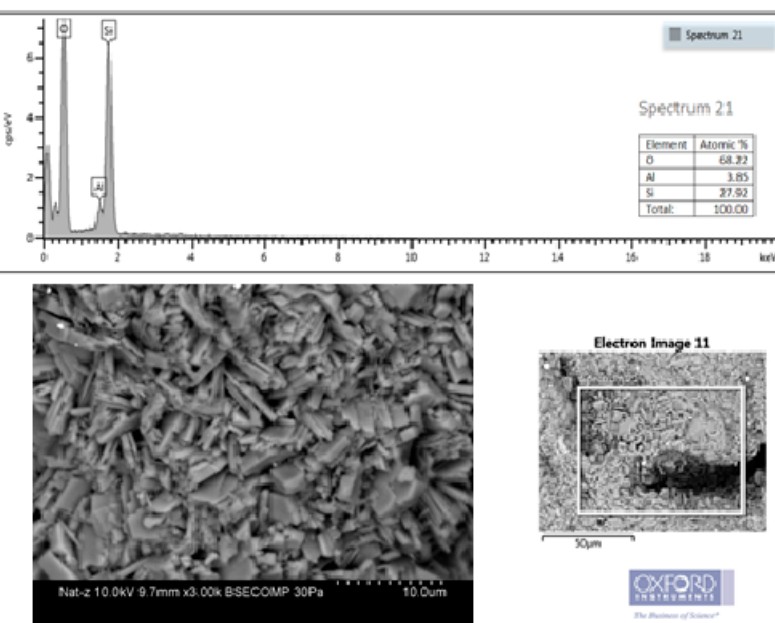

**Figure 3.** Electron microscopy images of the surface of the zeolite and its chemical composition.

### 3.3. Photo-Activity and Kinetics with the TiO$_2$/Ag Catalyst

Figure 4 shows the concentration profile of an NPEG photocatalytic degradation reaction at a concentration of 120 mg/L. It is possible to observe a decrease in the signal of the aromatic molecule (234 nm) corresponding to the π–π* transition, indicating the breakdown of chemical bonds and the degradation of NPEG molecules.

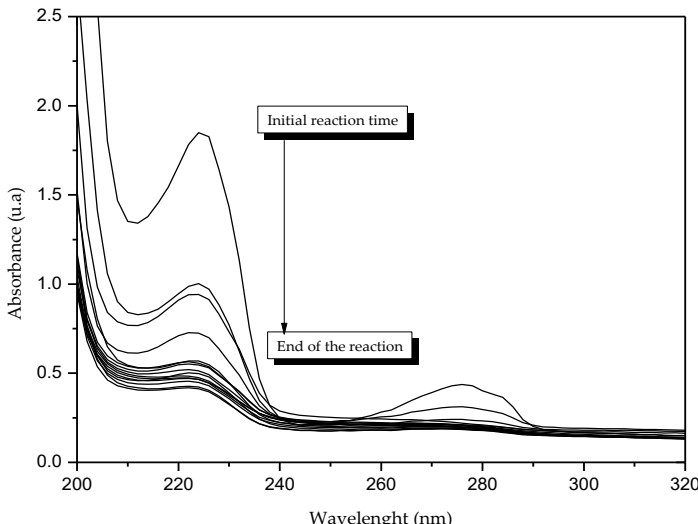

**Figure 4.** Degradation profile of nonylphenol polyethylene glycol (NPEG) measured by UV-vis spectroscopy (C = 120 mg/L, catalyst = 1 gr/L, oxygen volume = 100 cm$^3$, radiation = 4 UV lamps at λ = 365 nm, reaction volume: 200 mL).

The effect of the initial concentration of NPEG on the degradation reaction rate was investigated. Experiments with different initial NPEG concentrations between 20 and 120 mg/L (the concentration range was established on the basis of previous studies) were performed to establish the following reaction conditions: Catalyst concentration of 0.1 g of TiO$_2$/100 mL of solution, solution volume of 250 mL, and oxygen flow of 100 mL/min. In all cases, samples of the reaction mixture were taken for analysis with UV–vis and TOC. The results of these experiments (Figure 5) showed the changes in NPEG concentration and the total amount of organic carbon as a function of the reaction time.

The dependence of the initial concentration on the photocatalytic degradation rate of NPEG may be due to the fact that a degradation reaction occurred both in the TiO$_2$ molecules and in the solution. On the surface of TiO$_2$ molecules, a reaction occurred between the HO radicals generated at the active OH sites of the catalyst and NPEG molecules in the solution. In previous work, it has been shown that the experimental results of the photocatalytic oxidation of various organic compounds on TiO$_2$ under UV irradiation can be adjusted by the Langmuir–Hinshelwood (LH–HW) model [34]. The LH–HW model assumes a change of order in chemical reactions with concentration dependence, reaching a steady state at high concentrations of the substrate, so it is not possible to use this model at high concentrations.

When it is not possible to identify all organic intermediate products (OIP) from a chemical degradation reaction, the concentration can be obtained by means of a mass balance, based on total organic carbon measurements and measurements of the decrease in the concentration of the organic molecule, which allows the estimation of the overall production of OIP.

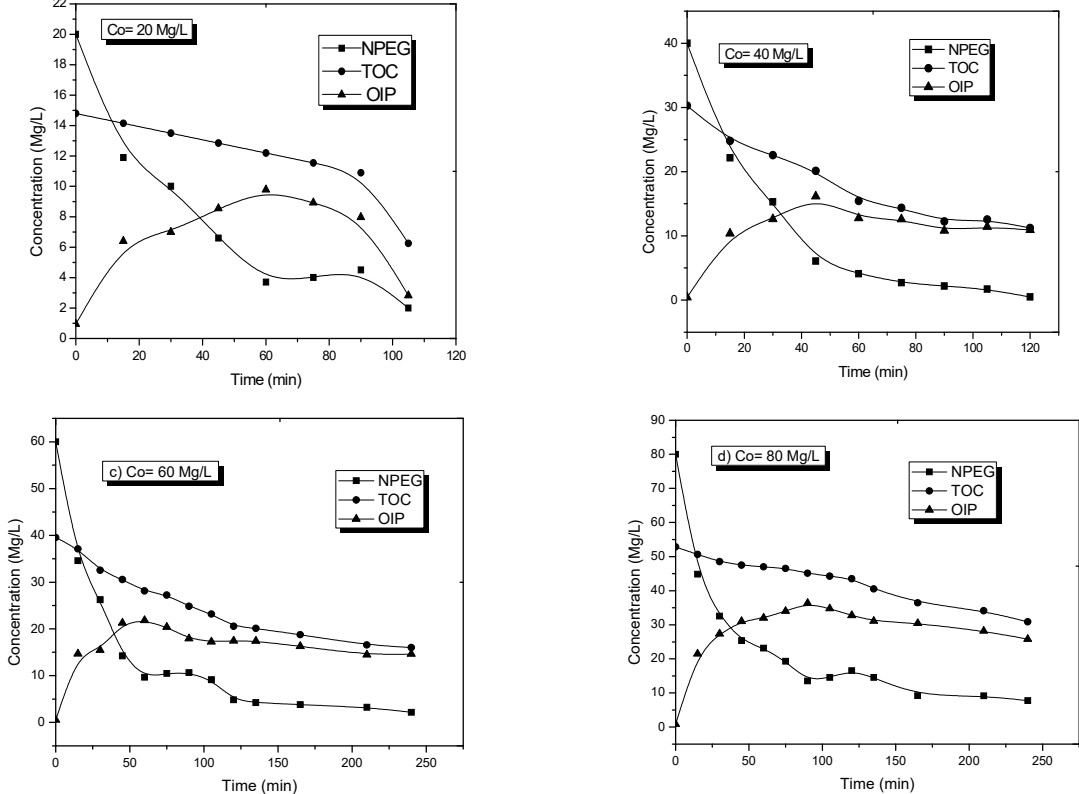

**Figure 5.** Nonylphenol polyethylene glycol (NPEG) concentration profile at different reaction times ((**a**) = 20 Mg/L, (**b**) = 40 Mg/L, (**c**) = 60 Mg/L, (**d**) = 80 Mg/L) showing the reaction behavior based on mass balances and total organic carbon profiles (TOC = total organic carbon, NPEG = concentration measured by UV-vis spectroscopy, OIP = organic intermediate products).

The chemical kinetics representing the decomposition of NPEG, which is a serial reaction, involves the estimation of four kinetic constants of rate and adsorption according to the LH–HW model:

$$\text{NPEG} \quad \underset{K_2}{\overset{K_1}{\rightarrow}} \quad \text{OIP} \quad \underset{K_3}{\overset{K_{11}}{\rightarrow}} \quad CO_2$$

The reaction rate equations for NPEG decomposition are represented by the following equation:

$$-r_{NPEG} = -\frac{dC_{NPEG}}{dt} = \frac{K_1 C_{NPEG}}{1 + K_2 C_{NPEG} + \sum K_i C_{OIP}} \qquad (1)$$

For the term $\sum K_i C_{OIP} = K_3 C_{OIP}$, it is assumed that all organic products of photocatalytic degradation of NPEG have the same adsorption constant and can be considered in an overall estimate; this term also includes the final concentration of all intermediate organic products; for that purpose, the mass balance based on TOC is useful and is calculated using the equation:

$$-r_{NPEG_{t=t}} = -\frac{dC_{NPEG}}{dT}\bigg|_{t=t} = \frac{K_1 C_{NPEG}}{1 + K_2 C_{NPEG} + K_3 C_{OIP}} \qquad (2)$$

The equation for the generation and the consumption of intermediate products is as follows:

$$r_{OIP} = \frac{K_1 C_{NPEG}}{1 + K_2 C_{NPEG} + K_3 C_{OIP}} - \frac{K_{11} C_{OIP}}{1 + K_2 C_{NPEG} + K_3 C_{OIP}} \qquad (3)$$

Due to the complexity of the differential Equations (2) and (3) for the estimation of the constants of $K_{11}$ and $K_3$, it was necessary to apply non-linear regression using the Quasi Newton

algorithm of Statistical package 7.1. The estimated values of the kinetic constants were as follows: ($K_1 = 0.025082$ min$^{-1}$, $K_2 = 0.00183$ MgL$^{-1}$ min$^{-1}$, $K_3 = 0.0107$ MgL$^{-1}$ min$^{-1}$, $K_{11} = 0.09430$ min$^{-1}$), with a 95% confidence level (alpha = 0.050).

The degradation behavior of NPEG, determined by UV-vis spectroscopy, and the mineralization behavior, determined by TOC, are shown in Figure 5. These two parameters were useful to obtain the average OIP curve using the equation:

$$TOC = C_{CARBON}NPEG + C_{CARBON}OIP \tag{4}$$

Figure 5 shows the degradation profiles of NPEG, the mineralization of organic compounds, as well as the estimation of OIP (Equation (4)).

Differential Equations (2) and (3) were solved with the Polymath software, using the kinetic constants calculated by the Runge Kutta's method.

The experimental data and model estimation using Equation (2) are shown in Figure 6; the model predicted the degradation of NPEG as well as the generation and consumption of OIP, based on Equation (3); this behavior is shown in Figure 7.

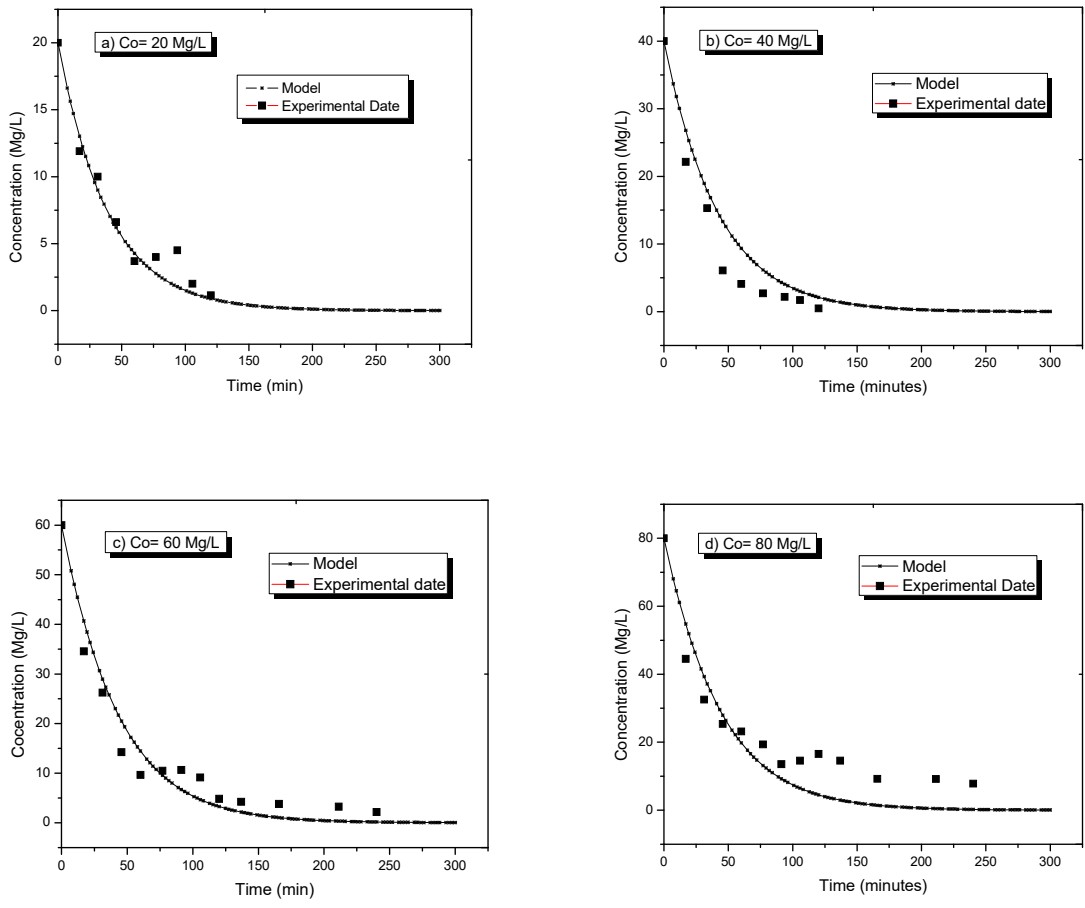

**Figure 6.** NPEG concentration profile ((**a**) = 20 Mg/L, (**b**) = 40 Mg/L, (**c**) = 60 Mg/L, (**d**) = 80 Mg/L) evaluated with the Langmuir–Hinshelwood (LH–HW) model that includes the estimation of the kinetic constants in Equation (2) as well as the experimental data.

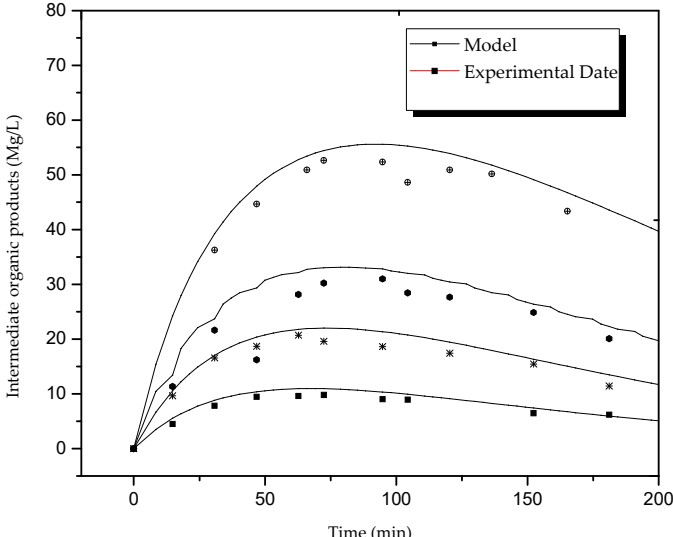

**Figure 7.** Profile of the distribution of OIP using the Langmuir–Hinshelwood (LH–HW) model with the estimation of the kinetic constants by Equation (3) and experimental data.

The experimental data and model estimation using Equation (2) are shown in Figure 6.

### 3.4. Adsorption Kinetics with Zeolites

Adsorption kinetics describe the rate of adsorption of the adsorbate on the adsorbent and determines the time at which a balance is reached. The adsorption rate describes how fast the concentration of the adsorbate changes per unit of time; however, the variation of the adsorbate concentrations over time is also relevant. The rate laws allow to calculate the rate of a reaction from the rate constant and the concentration of the reagents at any time during the process. This process is described with pseudo-second-order kinetic models (PSO).

Figure 8 shows the profile of the amount of NPEG in Mg/g adsorbed in time. It can be observed that the balance was reached after about 30 min, since the molecule is a surfactant and its recalcitrant character does not allow adsorption.

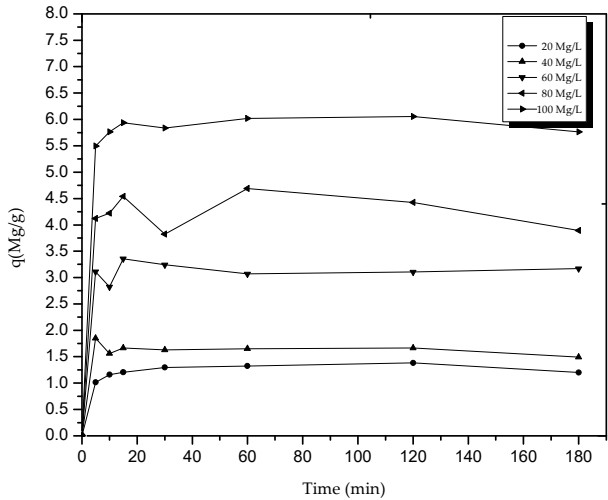

**Figure 8.** Profile of NPEG adsorption on natural zeolites at different concentrations.

The effect of the mass (Figure 9) of the adsorbent in terms of percentage of NPEG removal indicated that after the use of 0.4 g of zeolites, the removal capacity did not significantly increase, and the surfactant character of the molecule led to the formation of a film on the active sites preventing

their maximum adsorption; consequently, the balance constant increased with respect to the amount of mass due to the presence of more active sites, and the adsorption reached its balance quickly.

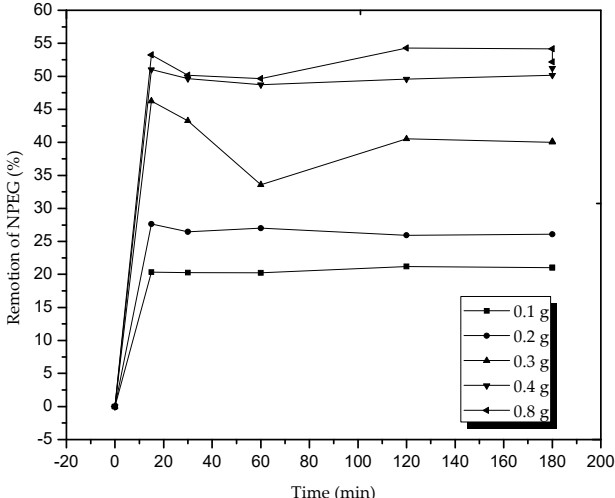

**Figure 9.** Effect of mass (in grams) on the removal capacity of zeolite, with an initial concentration of 100 Mg/L of NPEG.

The experimental data from the NPEG adsorption process were adjusted to the PSO linear model, which is based on adsorption in balance. If the adsorption is proportional to the number of active sites occupied in an adsorbent, then the equation is expressed as follows:

$$\frac{d_q}{d_t} = k_2 \left( q_e - q \right)^2$$

Integrating the equation with respect to time:

$$\frac{t}{q_t} = \frac{1}{k_2 q_e^2} + \frac{1}{q_e} t$$

where $q_t$ (Mg·g$^{-1}$) is the amount of adsorbed adsorbate at the established time, t (min) is the time of contact with the particle, $q_e$ is the adsorption capacity at balance (mg·g$^{-1}$), $k_2$ is the rate constant of the adsorption (mg·g$^{-1}$·min$^{-1}$), and $k_2 q_e^2$ (mg·g$^{-1}$·min$^{-1}$) is the initial rate of adsorption.

Table 1 shows the parameters of the kinetic models of NPEG absorbed on zeolite, adjusted to the PSO linear model and the PO (pseudo-first-order) for photocatalysis. It was observed that the kinetic constants had larger values for adsorption than for photocatalysis, which could be attributed to a faster achievement of balance. However, the photocatalytic process showed more extensive removal for NPEG. In adsorption higher removal percentages could not be achieved because the NPEG molecule is a surfactant and created a film that prevented the efficient absorption of NPEG on zeolite; another reason may be the lack of available active sites in the adsorbate. These removal processes are governed by different principles: While the adsorption process on zeolite does not modify the pollutant, in photocatalysis, the molecule of the pollutant is modified by changes in its chemical structure.

The determination of the adsorption balance parameters for NPEG was performed using the transformed linear equations of Freundlich, Langmuir, Temkin, as shown in Table 2. The linear models of Langmuir and Freundlich were not adjusted linearly. The $R^2$ correlation coefficients of the Temkin model were adjusted according to linear regression of the experimental data, indicating that the Temkin model adequately represented the experimental NPEG adsorption data.

**Table 1.** Values of the kinetic constants of the first-order adsorption models for photocatalysis.

| Kinetic Model | Pseudo Second Order (PSO) | | | Pseudo First Order (PPO) | | |
|---|---|---|---|---|---|---|
| Linear | $\frac{t}{q_t}=\frac{1}{k_2 q_e^2}+\frac{1}{q_e}t$ | | | $\ln\frac{Ca_t}{Ca_o}=-k\,t$ | | |
| Method | Adsorption | | | Photocatalysis | | |
| Ci (mg g$^{-1}$) | $k_2$(g min$^{-1}$ mg$^{-1}$) | $Q_e$ (mg g$^{-1}$) | $R^2$ | $k_1$ (s$^{-1}$) | $C_e$ (mg L-1) | $R^2$ |
| 20 | 0.260 | 1.416 | 0.999 | 0.022 | 0.700 | 0.977 |
| 40 | 0.472 | 1.697 | 0.999 | 0.015 | 1.300 | 0.888 |
| 60 | 0.275 | 3.171 | 0.999 | 0.010 | 6.200 | 0.911 |
| 80 | 0.066 | 4.835 | 1.000 | 0.102 | 8.470 | 0.888 |

**Table 2.** Adsorption balance parameters for NPEG according to the adsorption equations.

| Freundlich $q_e=k_F C_e^{\frac{1}{n}}$ | | | Langmuir $\frac{1}{q_e}=\frac{1}{q_m}+\frac{1}{q_m k_L C_e}$ | | | Temkin $q_e=B_T\ln k_T+B_T\ln c_e$ | | |
|---|---|---|---|---|---|---|---|---|
| $k_t$ mg$^{(1-n)}$ L$^n$ g$^{-1}$ | n | $R^2$ | $k_t$ L mg$^{-1}$ | Qm mg g$^{-1}$ | $R^2$ | $B_T$ | $k_t$ (L mg$^{-1}$) | $R^2$ |
| 0.169 | 1.201 | 0.808 | 0.015 | 3.179 | 0.757 | 3.867 | 0.127 | 0.994 |

The Langmuir isotherm is valid for molecules being adsorbed on a certain number of sites in fixed positions, with each site accepting only one molecule, and sites being available in the form of a monolayer; it also posits that lateral interactions between adsorbed molecules do not exist, and the adsorption rate is proportional to the concentration and the number of free sites [35]. On the other hand, Freundlich's isotherm is valid when there is no dissociation after adsorbtion on the surface and there is a complete absence of chemisorption [36]. The experimental kinetic results showed that our reaction conformed a PSO model, which supposes that chemisorption exists, therefore it could not be associated with a Freundlich isotherm, as corroborated by the linear adjustment. The Temkin's model [37] assumes that during adsorption, the heat of adsorption due to interactions with the adsorbate, decreases linearly with the recovery rate in relation to Gibbs' energy; our experimental results fitted this model.

Figure 10 shows the comparison profile of the two studied procedures based on the concentration removed. Both processes are usually suitable for the removal of NPEG; however, due to the recalcitrant nature of the molecule, it is usually more advisable to apply a process where there is chemical transformation. The conversion percentages are shown in Figure 11; the photocatalytic process allowed the greatest removal in the same conditions of time and initial concentration of the reactant.

Zeolites are usually efficient adsorbents of other aromatic derivatives. It appears that the hydrophilic character of the molecule defines the adsorption properties [30]. This is consistent with the behavior observed during adsorption; in fact, NPEG, due to its ethoxylated chains and its surfactant character, created foams in the first minutes of adsorption, which might block the adsorber's active sites.

In order to make comparisons and have a statistical support of the results, a random Block Design model was applied, $\mu + T_i + \beta_j + \varepsilon_{ij}$, with a 95% confidence level. The value of the test statistic was highly significant, indicating that there were statistically significant differences in the application of such treatments.

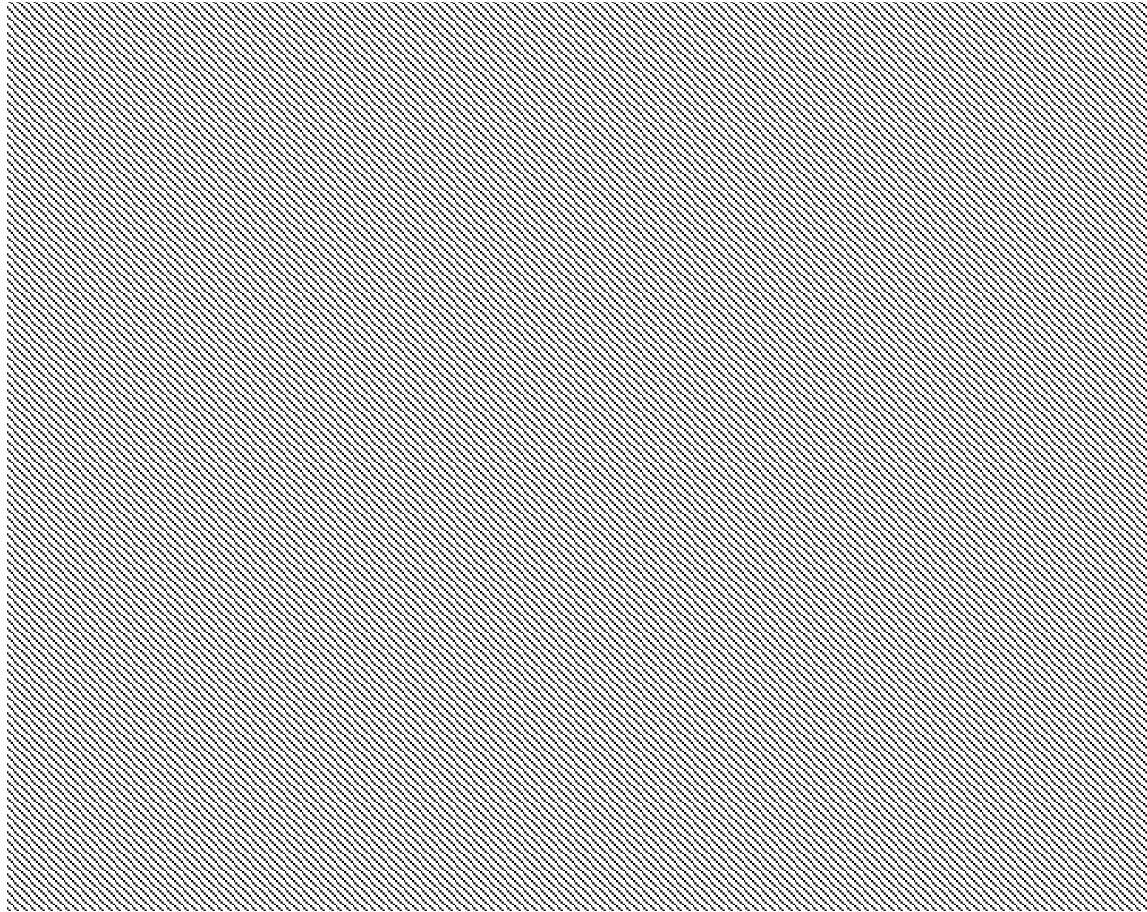

**Figure 10.** Profile of the removal capacity of the two treatments (photocatalysis and adsorption) evaluated according to the decrease in the concentration of NPEG (**a**) = 20 Mg/L, (**b**) = 40 Mg/L, (**c**) = 60 Mg/L, (**d**) = 80 Mg/L.

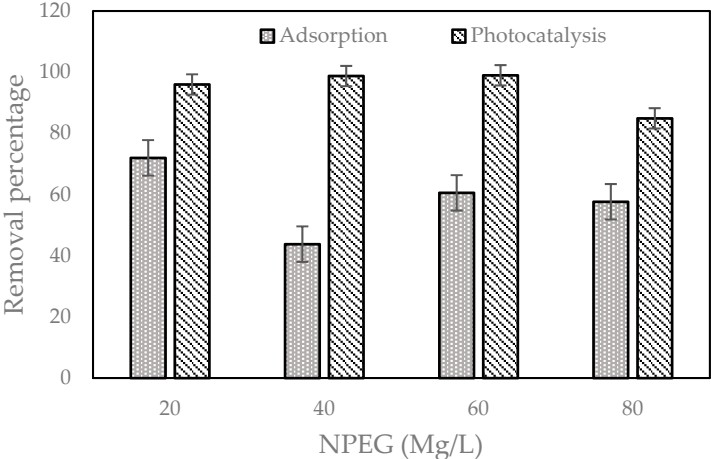

**Figure 11.** Percentages of removal by the two treatments (adsorption and photo-catalysis); based on the decrease in the initial concentration of NPEG, the photocatalytic process appeared more efficient than the adsorption process.

## 4. Conclusions

The two removal methods were efficient in removing NPEG from an aqueous (water) solution. However, due to the particularities of the molecule and its surfactant character that can affect the contact zone between two phase NPEG degradation in the aqueous environment was successful when

using the photocatalytic process; however hydrophobic character can lead to the formation of foam which creates a barrier on the adsorbent material, limiting adsorption.

NPEG creates a film on the adsorbent materials, quickly reaching equilibrium and a stationary state, which limits the transfer of mass and does not allow achieving removals greater than 60% of the mass. The values of the determination coefficients ($R^2$) showed that PSO models better predicted the experimental data of the photo-catalytic process. Temkin isotherms adequately predicted the experimental data of NPEG adsorption on the zeolite. In its natural state, the zeolite has proven to be a material capable of removing recalcitrant compounds; however, with the photo-catalytic process, it leads to the chemical transformation of molecules, generating intermediate products that, through the application of LH model equations, can be predicted in mass. The removal by photocatalysis exceeded 80%; the reaction kinetics showed that the removal of NPEG followed a PPO kinetics and that the reaction rate decreased as the concentration increased. For this reason, it is convenient to describe the kinetic reaction of the LH–HW model. These estimates are not sufficient to contemplate the phenomenon in its totality, and the estimation of the organic intermediate products was carried out with evaluating the experimental values through TOC and UV analyses. The numerical values obtained for the kinetic constant and the absorption constant of the intermediates were useful to predict the behavior of the reaction. Statistically, the results obtained by applying the two treatments were significantly different.

**Author Contributions:** C.A.U. and M.A. are the technical managers of the project; C.M.R., M.A.R.E., F.A.F. and D.C.L. collaborated in the writing of the manuscript.

**Funding:** The authors thank the National Council of Science and Technology of México (CONACYT) project 169404.

**Conflicts of Interest:** The authors declare no conflict of interest.

## Nomenclature

| | |
|---|---|
| $C_{NPEG}$ | NPEG concentration (mg/L) |
| $r_{NPEG}$ | Reaction rate |
| $K_1$ | NPEG degradation reaction rate constant ($min^{-1}$) |
| $K_2$ | NPEG adsorption constant (L/mg) |
| $K_3$ | Average adsorption constant (L/mg) |
| $K_{11}$ | Reaction rate constant for the mineralization of the organic products ($min^{-1}$) |
| OIP | Organic intermediate products |
| $K_3 C_{OIP}$ | Average absorption constant for the total concentration of the intermediates which formed. |
| $q_t$ (mg·g$^{-1}$) | Amount of adsorbate |
| $q_e$ | Adsorption capacity in equilibrium |

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
