# Peer review of "Removal of an Ethoxylated Alkylphenol by Adsorption on Zeolites and Photocatalysis with TiO2/Ag"

_processes, doi:10.3390/pr7120889_

Round 1

Reviewer 1 Report

This manuscript shows two techniques, photocatalytic and absorption process and presents the comparison of NPEG removal through these processes.
On technical merits, this is a very good and useful study.
However, there are a few missing points in the text, the introduction is not extensive enough, and quality of presentation of the obtained results is poor.
The obtained results are impactful and should be shared with the scientific community after the following comments are addressed adequately.

Correct the following issues with Figure 3:
-label three sub-figures as 3a, 3b, and 3c. Correct this for all other figures as well. No sub-figure is labeled as a, b, c, ...
-resolution of all three sub-figures is bad and details are not visible.
-Show clearly visible scalebar for SEM pictures.

Fig 5, the last sub-figure shows the wrong concentration value. Author must carefully read the paper before the resubmission.

It is surprising that author has not mentioned any other examples outside current system that have used surfactant nature of the materials for similar applications, which leads to an incomplete discussion. Perspective readers must know that this property has been very useful and widely used and it is not limited to NPEG alone. Author should add at least a few sentences to the introduction to put light on this subject. Following is an example with such a few studies:

"One of the reasons behind the high efficiency of the removal of NPEG is its surfactant nature. This excellent colloidal phenomenon has also been successfully used for other material systems in the past few decades. It has lead to purification, control, filtration, and separation of many other substances, e.g metal spherical particles and nanorods (ref.-Lisiecki 2000, Lisiecki 1996), non-metal nano-particles and 2-dimensional materials (ref- Kumar 2016, Dickinson 2018), etc.

Ref. for metal nanorods and nanoparticles

-Lisiecki, 2000, Phys. Rev. B 61, 4968. (https://doi.org/10.1103/PhysRevB.61.4968)
-Lisiecki, J. Phys. Chem.1996, 100, 10, 4160-4166, (https://pubs.acs.org/doi/10.1021/jp9523837)

Ref. for non-metal nanoparticles and 2-dimensional materials

-Kumar, J. Colloid Interface Sci. 2017, 493:365-370. (DOI: 10.1016/j.jcis.2017.01.043)

-Dickinson, Nanoscale, 2018,10, 14441-14447.(https://doi.org/10.1039/C8NR01725E)"

Section 3.2, EDS measurements:
EDS quantitative elemental analysis for lower atomic weight elements, e.g. H, and O in the present context, is not useful. However, for hight weights, it is good. Either author must avoid EDS for H and O or clearly mention in the text that "the values might not be accurate for H and O" and are for qualitative purposes only. If however, the author wants to use EDS, the correct method should be followed. One of such references author might follow (or any other) is:
J Mater Sci. 2015; 50(2): 493–518. (doi: 10.1007/s10853-014-8685-2).

Line 257, How the concentration value range is chosen, e.g 20, 40..... How is this limit (20 or 100) decided?

Does the model (LH-HW) follow any NPEG concentration limits, why or why not?

Figure 11 histogram data needs error bars. A comparison of scientific data through histograms without error bars is not meaningful.

Author Response

Point 1: revisor

Correct the following issues with Figure 3:
-label three sub-figures as 3a, 3b, and 3c. Correct this for all other figures as well. No sub-figure is labeled as a, b, c, ...
-resolution of all three sub-figures is bad and details are not visible.
-Show clearly visible scalebar for SEM pictures.

reply:  It was not possible to modify this figure (3) I do not have the software data. Only as an image.

point 2: revisor 

Fig 5, the last sub-figure shows the wrong concentration value. Author must carefully read the paper before the resubmission.

reply: 

Graph 5 was corrected and identified as a), b)

point 3: revisor

It is surprising that author has not mentioned any other examples outside current system that have used surfactant nature of the materials for similar applications, which leads to an incomplete discussion. Perspective readers must know that this property has been very useful and widely used and it is not limited to NPEG alone. Author should add at least a few sentences to the introduction to put light on this subject. Following is an example with such a few studies:

reply: a paragraph was added in introduction that highlights the importance of surfactants and highlighted the importance in the conclusions

point 4: revisor
EDS quantitative elemental analysis for lower atomic weight elements, e.g. H, and O in the present context, is not useful. However, for hight weights, it is good. Either author must avoid EDS for H and O or clearly mention in the text that "the values might not be accurate for H and O" and are for qualitative purposes only. If however, the author wants to use EDS, the correct method should be followed. 

reply: the correction is made mentioning that the data is not accurate for oxygen, but for the elements of greater weight the data is approximate

point 5: revisor

Line 257, How the concentration value range is chosen, e.g 20, 40..... How is this limit (20 or 100) decided?

Reply: 

concentrations are based on previous studies,

Does the model (LH-HW) follow any NPEG concentration limits, why or why not?

reply: The active sites of the catalyst become saturated when using high concentrations and the results show us stationary behavior. It is not appropriate to use high concentrations

Figure 11 histogram data needs error bars. A comparison of scientific data through histograms without error bars is not meaningful.

reply: 

error bars were added

Reviewer 2 Report

                The manuscript concerns an interesting and important problem: removal of ethoxylated alkylophenols from water. Ethoxylated alkylophenols are not very toxic themselves , but their degradation produces harmful materials. The authors used two kinds of materials: titanium dioxide doped with silver and zeolite of clinoptilolite type. It should be noted, that clinoptilolite is non-expensive zeolite, the natural deposits of which are very rich just in Mexico.

                Both kinds of materials were characterized before of catalytic or adsorption studies. Both photocatalytic experiments and XRD and SEM microscopic studies were realized. The analysis of kinetic data of photodegradation and the analysis of adsorption isotherms were done. Various adsorption isotherms were compared and the Temkin isotherm was found to be the best.

                Finally, the process of photocatalytic degradation was found to more efficient than adsorption.

                I propose to accept the manuscript with only small modification. I propose only to improve language (some elements of Spanish are seen). I propose also to correct some symbols: instead of [AlO4]-5 it should be [AlO4]- and instead of [SiO4]‑4 it should be [SiO4].

                I have also an advice to Authors. I advice to try use other zeolite: faujasite, which is large pore zeolite and is also not expensive one. Large pore zeolite may be more efficient for adsorption of bulky ethoxylated alkylophenols molecules.

Author Response

point 1: revisor 

  I propose to accept the manuscript with only small modification. I propose only to improve language (some elements of Spanish are seen). I propose also to correct some symbols: instead of [AlO4]-5 it should be [AlO4]- and instead of [SiO4]‑4 it should be [SiO4].

reply: The corrections suggested in the nomenclature were made

point 2: revisor

  I have also an advice to Authors. I advice to try use other zeolite: faujasite, which is large pore zeolite and is also not expensive one. Large pore zeolite may be more efficient for adsorption of bulky ethoxylated alkylophenols molecules

reply:  

t is an excellent recommendation that will be taken into account. We welcome your suggestions.

point 3: revisor

 I propose to accept the manuscript with only small modification.
I propose only to improve language (some elements of Spanish are seen).
I propose also to correct some symbols: instead of [AlO4]-5 
it should be [AlO4]- and instead of [SiO4]‑4 it should be [SiO4].

reply:
The writing was revised so as not to leave data in Spanish and
the nomenclature was corrected.

Round 2

Reviewer 1 Report

Clear SEM scalebar is still needed. Rest, the author has made required changes and the paragraph about EDX analysis is satisfactory.